# Delivery Room Care for Premature Infants Born after Less than 25 Weeks’ Gestation—A Narrative Review

**DOI:** 10.3390/children8100882

**Published:** 2021-10-02

**Authors:** Bernhard Schwaberger, Berndt Urlesberger, Georg M. Schmölzer

**Affiliations:** 1Division of Neonatology, Department of Pediatrics and Adolescent Medicine, Medical University of Graz, 8036 Graz, Austria; bernhard.schwaberger@medunigraz.at (B.S.); berndt.urlesberger@medunigraz.at (B.U.); 2Centre for the Studies of Asphyxia and Resuscitation, Neonatal Research Unit, Royal Alexandra Hospital, Edmonton, AB T5H 3V9, Canada; 3Department of Pediatrics, Faculty of Medicine and Dentistry, University of Alberta, Edmonton, AB T5H 3V9, Canada

**Keywords:** threshold of viability, decision making, active management, postnatal stabilization, umbilical cord management, respiratory support, cardio-circulatory support, premature infants born after <25 weeks’ gestation

## Abstract

Premature infants born after less than 25 weeks’ gestation are particularly vulnerable at birth and stabilization in the delivery room (DR) is challenging. After birth, infants born after <25 weeks’ gestation develop respiratory and hemodynamic instability due to their immature physiology and anatomy. Successful stabilization at birth has the potential to reduce morbidities and mortalities, while suboptimal DR care could increase long-term sequelae. This article reviews current neonatal resuscitation guidelines and addresses challenges during DR stabilization in extremely premature infants born after <25 weeks’ gestation at the threshold of viability.

## 1. Introduction

The impending birth of periviable infants born after 22 + 0 to 24 + 6 weeks’ gestation is associated with anxiety and uncertainty for family members and healthcare teams [1]. Parental beliefs and values need to be incorporated into the decision-making process to provide the optimal approach and outcome for the parent–infant dyad. These options include active or comfort (palliative) care. The decision of active management leaves clinicians in a challenging position. Despite the advances that have been achieved in perinatal and neonatal care, there is a lack of strong evidence in delivery room (DR) studies which have included infants at 22 + 0 to 24 + 6 weeks’ gestation [2].

In recent decades, gestational age has continuously shifted lower. However, the stabilization of premature infants born after <25 weeks’ gestation in the DR remains challenging [3]. The current neonatal resuscitation guidelines are designed for term and premature infants born after <32 weeks’ gestation [4,5,6,7], thereby potentially not providing the optimal DR management for infants born after <25 weeks’ gestation. The aim of this narrative review was to address the available evidence including current recommendations regarding DR stabilization and resuscitation in premature infants born after <25 weeks’ gestation with a focus on respiratory and cardio-circulatory support.

## 2. Decision Making at <25 Weeks’ Gestation

Decision making about active or comfort (palliative) care in infants to be delivered between 22 + 0 to 24 + 6 weeks’ gestation is challenging. Gestational age (despite the uncertainty of the accuracy of an infant’s gestation determined by fetal ultrasound) is used as the main determinant for decision-making due to its association with outcomes [8,9]. Current recommendations vary widely and decision making should use a guided approach including parents and care-givers at each hospital. Factors that might influence the decision-making process include birth weight, multifetal pregnancy, sex [10], presence of intra-amniotic infection [11] or the presence of congenital anomalies. In addition, antepartum corticosteroid [12,13] and magnesium sulfate [14] administration could influence outcomes. Survival predicting tools such as the Neonatal Research Network Extremely Preterm Birth Outcome Model are widely used for prediction by care-givers and thereby may support the decision-making process [15]. However, systematic analyses investigating the impact on neonatal outcome by incorporating survival predicting tools into the decision-making process are lacking.

Survival rates among periviable infants born after 22 + 0 to 24 + 6 weeks’ gestation increase for deliveries that occur in hospitals with NICUs that have both a high level of care and a high volume of such patients [16]. Different perinatal management among hospitals and countries may have a further impact on outcome. In centers with a more proactive perinatal resuscitation strategy in premature infants born after <25 weeks’ gestation, the number of live births increased and survival rates improved when compared to those with a more selective approach, potentially resulting in increased morbidity among survivors [17,18]. Most importantly, there is an open discussion about the parents’ wishes, the capability of the perinatal center and NICU and their outcomes to provide the best decision-making options for the parents.

## 3. Umbilical Cord Clamping

### 3.1. Delayed Cord Clamping

Immediate cord clamping (ICC) has been used for several decades. However, evidence suggests that ICC might cause an acute reduction in left atrial filling, leading to an abrupt drop in left ventricular output [19]. In contrast, delayed cord clamping (DCC) might improve blood pressure stability and placental transfusions in premature infants [19,20]. A meta-analysis including 3514 premature infants born after <34 + 0 weeks’ gestation from 23 studies reported that DCC may improve neonatal survival or reduce neonatal mortality with a survival risk ratio of 1.02, a 95% confidence interval (CI) of 1.00–1.04 with a number needed to benefit: 50, 95% CI: 25 to no benefit [21]. However, most studies included in the meta-analysis only enrolled infants born between 32 and 34 weeks’ gestation. The trial by Tarnow-Mordi et al. included 518 premature infants born after < 27 + 0 weeks’ gestation and reported no differences in the composite of death or major morbidity (*p* = 0.23) [22]. While the evidence is limited in premature infants born after <25 weeks, recent guidelines recommend that DCC be performed irrespective of gestational age [4,5,6].

### 3.2. Umbilical Cord Milking

Intact umbilical cord milking (I-UCM) has been advocated as an alternative to DCC, particularly in infants who do not breathe at birth [19]. Animal studies reported that I-UCM promotes placental transfusion, however, this causes large fluctuations in mean carotid artery pressures and carotid artery blood flows (with each milking along 10 cm of cord, carotid artery blood flow increased and decreased by 15% ± 2% and 8% ± 1%, respectively) [23]. A pilot trial comparing I-UCM with DCC reported higher superior vena cava flow and right ventricular output during the first 12 h of life [24] with higher cognitive composite score (100 ± 13 vs. 95 ± 12, *p* = 0.031) and language composite score (93 ± 15 vs. 87 ± 13, *p* = 0.013) at 22–26 months of corrected age compared with those randomized to DCC [25]. However, a recent large randomized trial comparing I-UCM with DCC reported a significant increase in the rates of severe intraventricular hemorrhage (IVH) after I-UCM in a subgroup of 182 premature infants born after <28 weeks’ gestation (20 vs. 5; *p* = 0.02) [26]. This led to an early termination of the trial, and currently I-UCM is not recommended in premature infants born after <28 weeks’ gestation [6].

An alternative approach of umbilical cord milking is cut-umbilical cord milking (C-UCM), which is performed by early cord clamping and retaining a long segment of the umbilical cord that then can be milked while initiating ventilation [27]. In extremely low-birth-weight (ELBW) infants, C-UCM increases the mean blood volume by 17.7 (±5.5) mL/kg birthweight per 30 cm of umbilical cord [28] and has similar effects on placental transfusion and the need for red blood cell transfusions compared to I-UCM [27]. In premature infants, C-UCM may increase peak hematocrit within 24 h after birth compared to ECC, but the study population only included six (10%) ELBW infants [21,29]. Prospective trials in premature infants born after <25 weeks’ gestation are warranted to prove the safety and effects of C-UCM on short- and long-term outcomes.

### 3.3. Intact Cord Resuscitation

Animal models demonstrated that hemodynamic transition improves when respiratory support is provided while preterm lambs remain attached to the cord (i.e., intact cord resuscitation). Randomized trials have compared intact cord resuscitation with ECC [30], DCC [31,32], or I-UCM [33], and reported the feasibility of 59–100% [33,34,35,36]. However, these trials have not reported improved outcomes and did not include a sufficient number of ELBW infants.

## 4. Temperature Control

Thermal care to maintain body temperature between 36.5 and 37.5 °C is crucial to reduce morbidity and mortality in premature infants [37,38]. Admission temperature is a strong prognostic factor in low-birth-weight infants [39,40]. Placed under a preheated radiant heater, the premature infant should be completely covered with polyethylene wrapping (apart from the face) without prior drying [6,7,37,38]. The temperature should be regularly monitored after birth to prevent hypo- and hyperthermia [6]. Further interventions might be beneficial, including the use of warmed humidified gases in infants receiving respiratory support, increased room temperature > 25 °C, warm dry blankets, thermal mattress and a head cap [6,7,37,38,41,42].

## 5. Respiratory Support in the Delivery Room

### 5.1. Spontaneous Breathing

Approximately 80% of ELBW infants initiate spontaneous breathing at birth. However, all of them require CPAP and/or PPV after birth due to their weak respiratory drive, low respiratory muscle strength and lung immaturity [43]. The presence of spontaneous breathing seems to be essential, since larynx adduction during apnea can impede gas from entering the lungs during non-invasive respiratory support [44]. Thus, the larynx in premature infants needs to be opened after birth to establish lung aeration. This occurs in newborns who achieve spontaneous breathing [44,45].

Tactile maneuvers might stimulate breathing and thereby improve oxygenation in premature infants [32,46,47]. Katheria et al. have shown that gentle tactile stimulation in premature infants during DCC promotes the establishment of spontaneous breathing and provides a similar placental transfusion compared to CPAP and/or PPV during DCC [32]. Baik-Schneditz et al. reported that oxygenation improved after tactile stimulation (before 61.9 (53.1–76.0) versus after stimulation 67.8 (58.1–77.1), *p* < 0.001) in late premature infants [47]. Repetitive tactile stimulation compared with standard stimulation (based on clinical indication) in premature infants 27–32 weeks’ gestation resulted in significantly improved oxygenation with a lower fraction of inspired oxygen (FiO_2_), but did not result in differences in respiratory effort between groups [46]. Several additional studies reported on tactile stimulation in premature infants >30 weeks’ gestation. However, only the study by Katheria et al. included premature infants at <25 weeks’ gestation. Future trials are required to assess whether tactile maneuvers improve respiratory function as well as to identify the best tactile stimulation strategy (i.e., stimulation area, duration, frequency, etc.) with a special focus on premature infants born after <25 weeks’ gestation [48].

Another strategy to improve the breathing effort after birth in premature infants is the postnatal administration of intravenous caffeine. Caffeine administration was associated with increased diaphragmatic activity and tidal volume within five minutes of its administration in infants born between 26 and 34 weeks’ gestation [49]. A randomized trial comparing caffeine with no caffeine in the DR reported an increased minute ventilation in spontaneous breathing premature infants 24–30 weeks’ gestation [50]. However, the use of caffeine in the DR needs further exploration in terms of clinical outcomes.

More controversial might be the effect of oxygen on the initiation of spontaneous breathing in premature infants [51]. A randomized controlled trial compared an initial FiO_2_ of 1.0 versus 0.3 and reported significantly higher breathing efforts, improved oxygenation, and a shorter duration of PPV with a FiO_2_ of 1.0, while an increased risk of hyperoxemia or oxidative stress was not observed [51]. Additionally, higher initial FiO_2_ resulted in improved tonic diaphragmatic activity [42]. This strategy requires further studies, as current neonatal resuscitation guidelines [4,5,6,7] recommend low initial FiO_2_ as discussed in Section 5.3. Oxygen titration.

Oropharyngeal or nasopharyngeal suctioning immediately after birth may delay the onset of spontaneous breathing, and therefore, should only be considered in the case of visible airway obstruction [7,52]. Suction should be performed in the case of airway obstruction during ventilation [5,6].

The presence of spontaneous breathing after birth has an enormous effect on successful lung aeration and on the establishment of functional residual capacity [44,45]. It can be stimulated by strategies such as tactile stimulation or intravenous postnatal caffeine administration and may be impeded by hypoxia or suctioning [5,6,7,32,46,47,48,49,50,51,52].

### 5.2. Initial Respiratory Support

The goal of respiratory support in the DR is to create a functional residual capacity, establish gas exchange, and initiate spontaneous breathing while minimizing acute lung injury [4]. Current neonatal resuscitation guidelines recommend a peak inflation pressure of 20–25 cmH_2_O [4,5,6]. However, a peak inflation pressure of 20 cmH_2_O might be too low to effectively recruit the lungs in extremely premature infants [53,54,55] due to their small airways and high airway resistance following the Hagen–Poiseuille equation. The European resuscitation guidelines recommend “five inflations maintaining the inflation pressure for up to 2–3 s” [6], while the North American resuscitation guidelines recommend PPV [5]. Several trials comparing sustained inflations, in which an inflating pressure is held for a prolonged duration greater than 5 s with PPV demonstrated no differences in the primary outcome of death before hospital discharge or secondary outcome parameters [56,57,58]. The SAIL trial, which enrolled 426 infants (23 to 26 weeks’ gestation), was terminated early due to a higher rate of death within the first 48 h of life in the sustained inflation group. The subgroup of premature infants born between 23 and 24 weeks’ gestation was predominantly affected [59].

During respiratory support, applying a face mask might induce a trigeminocardiac reflex provoking apnoea and bradycardia in a large proportion (54%) of premature infants [60]. This may compromise the capacity of premature infants to breathe and hereby may increase the necessity of applying PPV. Interestingly, this effect was inversely associated with gestational age (OR = 1.421 (1.281–1.583), *p* < 0.001) [60]. Applying bi-nasal prongs compared to a face mask for initial respiratory support did not result in a different incidence of apnoea among ELBW infants. However, the apnoea incidence was rather high for both interfaces (43/65 (66%) versus 46/65 (71%), *p* = 0.70) [61,62].

In addition to face masks and bi-nasal prongs, nasal tubes might be equivalent alternative interfaces for PPV at birth in ELBW infants [63]. Other supraglottic airways such as oropharyngeal airways or laryngeal masks are currently not recommended in premature infants born after <25 weeks’ gestation, since oropharyngeal airways significantly increased the incidence of airway obstruction and appropriately sized laryngeal masks are not available for those patients [6,64,65]. Administering ventilation at birth with a T-piece resuscitator compared with a self-inflating bag reduces the duration of PPV and the risk of bronchopulmonary dysplasia [66].

### 5.3. Oxygen Titration

Premature infants are highly susceptible to oxygen toxicity due to a deficient anti-oxidative capacity, exacerbating morbidities such as bronchopulmonary dysplasia (BPD), retinopathy of prematurity, IVH and necrotizing enterocolitis [1,67]. Alternatively, hypoxemia during postnatal transition is a significant risk factor for brain injury in the premature infant such as IVH and periventricular leukomalacia [68]. Blended oxygen and pulse oximetry monitoring enables titrating oxygen delivery to reduce hyperoxemia and hypoxemia during stabilization after birth [4,5].

A meta-analysis of seven trials (<500 total patients) in premature infants born after ≤28 weeks’ gestation reported no differences in mortality, major morbidity, or neurodevelopmental outcomes when respiratory support was started with low (0.21–0.3) compared with higher oxygen (0.6–1) [69]. Recent resuscitation guidelines recommend an initial FiO_2_ of 0.21–0.3 [5,7] or 0.3 [4,6] for premature infants born after <28 weeks’ gestation, which reflects a preference to prevent exposure to additional oxygen beyond what is necessary to achieve oxygen saturation targets. However, a subgroup analysis of premature infants born after <25 weeks’ gestation is not available. Since the current recommendations for oxygen therapy in premature infants come from low-quality evidence, the optimal FiO_2_ to initiate respiratory support in those patients after birth remains a hot topic for future research [67]. In particular, studies of intermediate oxygen concentrations (e.g., 0.4–0.5%) to initiate resuscitation in premature infants are urgently required.

During postnatal stabilization, FiO_2_ adjustments should be guided by pulse oximetry. Oxygen titration should be attempted every 30 s to meet predefined oxygen saturation targets [70]. Not achieving an oxygen saturation of at least 80% by 5 min after birth is associated with adverse outcomes including major IVH [71]. A meta-analysis with data from eight trials demonstrated that, in the subgroup of premature infants born after <25 weeks’ gestation, only 14 of 46 (30%) subjects exceeded the oxygen saturation target of 80% by 5 min after birth [71]. It remains uncertain whether this was due to insufficient oxygen administration or poor pulmonary function. Optimal oxygen saturation, optimal oxygen titration in the DR, and the use of near-infrared spectroscopy during the stabilization of premature infants born after <25 weeks’ gestation needs further research with the aim to generate higher quality evidence adequately powered for neurodevelopmental outcomes [70,72].

### 5.4. Continuous Positive Airway Pressure

Spontaneously breathing premature infants demonstrating respiratory distress should receive respiratory support by continuous positive airway pressure (CPAP) with at least 5–6 cmH_2_O via either a face mask or nasal prongs rather than endotracheal intubation and mechanical ventilation [4,6,73,74]. A meta-analysis by Schmölzer et al. [74] included four trials with 2782 infants born after <30 weeks’ gestation. The pooled analysis showed a significant benefit for the combined outcome of death or BPD, or both, at 36 weeks corrected age for babies treated with nasal CPAP (relative risk 0.90 (95% confidence interval 0.83–0.98, risk difference −0.04 (95% confidence interval from −0.08 to −0.00), the number needed to treat of 25). However, only the SUPPORT trial [75] included premature infants born after <25 weeks’ gestation, and none included infants born after <24 weeks’ gestation. In a subgroup of premature infants born between 24 and 25 weeks’ gestation, rates of death during hospitalization and at 36 weeks were significantly lower in the CPAP group compared to the mechanical ventilation one [75].

The optimal CPAP levels for initiating respiratory support in the DR in spontaneously breathing premature infants born after <25 weeks’ gestation should be addressed in future trials. Interestingly, a recent animal study has shown that higher initial CPAP levels (15 cmH_2_O) compared to 5 cmH_2_O resulted in higher pulmonary blood flow levels when applied from birth, without causing pulmonary overexpansion, cardiovascular compromise or increased risk for IVH [76]. Indeed, while higher PEEP levels during mechanical ventilation enhance lung aeration and oxygenation and reduce intubation rates, they reduce pulmonary blood flow and may increase the risk of pneumothorax [77,78]. Nevertheless, physiology might be immediately different after birth when the high resistance lungs need to be cleared from fluids. A new, promising, ‘physiological’ approach that needs further investigation is a dynamic (high) CPAP level during the first minutes after birth (e.g., 15 decreasing to 8 cmH_2_O at ~2 cmH_2_O/min) which takes into account the changes in lung function during stabilization in the DR [76].

### 5.5. Surfactant Administration

Surfactant administration to treat the respiratory distress syndrome can improve survival in premature infants by preventing alveoli collapse and increasing lung compliance [4,73]. There are different methods of surfactant administration including (i) less invasive surfactant administration (LISA); (ii) minimally invasive surfactant therapy (MIST); or (iii) intubate–surfactant–extubate (INSURE) [79]. Their overall aim is to provide surfactant as early as possible once it is deemed necessary while avoiding invasive ventilation [4].

A meta-analysis including six trials of premature infants born between 23 and 34 weeks’ gestation reported a reduced composite outcome of death or BPD at 36 weeks’ gestation with LISA compared to INSURE [80]. One trial included 211 very premature infants born between 23 and <27 weeks’ gestation but was unable to prove an improvement in BPD-free survival [81]. However, in this trial, LISA was associated with lower rates of severe IVH [81]. A further systematic review compared mechanical ventilation and different non-invasive strategies, and LISA was associated with the lowest likelihood of death or BPD at 36 weeks [82]. LISA has been successfully used in infants born as prematurely as 22 weeks [79]. Some centers routinely use LISA in the DR in infants born after <25 weeks’ gestation when they have increased oxygen requirements or respiratory distress. However, this ‘quasi-prophylactic’ approach has not been studied in any trials [79]. There are limited data on neurodevelopmental outcomes at 18–24 months in premature infants born after <32 weeks comparing LISA and INSURE. However, no differences in respiratory morbidities, sensorineural deficits, or adverse neurodevelopmental outcome have been reported [83,84].

Alternatively, INSURE comprises endotracheal intubation, surfactant administration followed by a (brief) period of PPV, and subsequent early extubation [85]. INSURE, compared with delayed selective surfactant administration and ongoing mechanical ventilation, reduces the need for and duration of mechanical ventilation [86]. However, none of the trials included premature infants born after <25 weeks’ gestation. The need for sedation and associated adverse effects such as bradycardia or hypotension, and the potential harm associated with endotracheal intubation and brief periods of mechanical ventilation are concerns of the INSURE technique [73,87].

Endotracheal intubation and the subsequent invasive ventilation should be reserved for premature infants not responding to PPV during DR stabilization, and they should then receive surfactant via the endotracheal tube [4].

## 6. Cardio-Circulatory Support

### 6.1. Chest Compressions and Epinephrine Administration

None of the international resuscitation guidelines endorse withholding chest compressions and/or epinephrine administration from the subgroup of premature infants born after <25 weeks’ gestation [5,6,7], although there are concerns over whether extensive cardiopulmonary resuscitation should be performed in such premature infants, as both have been associated with high mortality and neurodevelopmental impairment rates [88,89,90,91]. A Canadian cohort study included 190 ELBW infants with a mean (SD) gestational age of 25.4 (1.7) weeks and 29% of subjects born after ≤24 weeks’ gestation who received chest compressions and/or epinephrine administration in the DR. Overall, 60% survived with 78% of the survivors not severely impaired at 18–24 months corrected age [91]. A Vermont Oxford Network study reported a higher survival rate in a subgroup of premature infants weighing between 401 and 500 g who received extensive cardiopulmonary resuscitation compared to neonates who did not receive chest compressions and/or epinephrine administration. These findings were attributed to a more aggressive ventilation and resuscitation approach in the investigated cohort [92]. In the absence of evidence to justify a different approach in ELBW infants, the standard newborn resuscitation algorithms, including chest compressions and/or epinephrine administration, should also be used in premature infants born after <25 weeks’ gestation, if advanced resuscitation was considered appropriate prior to birth [93].

### 6.2. Vascular Access

The current neonatal resuscitation guidelines recommend vascular access if infants require chest compressions or fluid administration [6]. Vascular access could be achieved with either umbilical venous catheterization, peripheral, or intraosseous access [6,7]. While umbilical venous catheterization is most commonly used, the success rates and adverse effects attributable to emergency umbilical venous catheterization are unknown. Peripheral venous access has been described in premature infants (mean gestational age of 31 ± 2 weeks) with a high success rate at 5 (IQR 4–9) minutes after birth [94]. However, the study did not include premature infants born after <25 weeks’ gestation, which might result in lower success rates.

More recently, intraosseous access has been advocated as an alternative to umbilical venous catheterization [6]. A total of 80 newborns have been described in case series [95,96], with varying success rates and severe complications such as bone fracture, osteomyelitis, compartment syndrome, and amputation [95]. There have been reports on the successful use of a Cook needle (Cook Medical, Bloomington, IN, USA) or butterfly needle in five infants at 25 weeks’ gestation and a lowest birth weight of 515 g [97,98], but none in premature infants born after <25 weeks’ gestation. While there are currently no data to support intraosseous lines in premature infants born after <25 weeks’ gestation, it might be considered during neonatal resuscitation if other access routes have failed.

## 7. Conclusions

Once the decision to provide full life support in premature infants born after <25 weeks’ gestation is made, the active management of such infants remains challenging due to the fact that evidence for periviable neonates is lacking and guidelines are mainly based on expert consensus, physiologic plausibility, as well as data derived from more mature low-gestational-age infants. In recent years, there has been growing evidence for providing gentle, less invasive support in the DR to reduce mortality and short- and long-term morbidities. Further research into all the aspects of stabilization and resuscitation focusing on the subgroup of premature infants born <25 weeks’ gestation is urgently warranted.

## Data Availability

This is a review, and there was no original data.

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
