# Peer review of "Delivery Room Care for Premature Infants Born after Less than 25 Weeks’ Gestation—A Narrative Review"

_children, 2021, doi:10.3390/children8100882_

Round 1

Reviewer 1 Report

The authors provide a review of the available evidence on resuscitating preterm infants <25 weeks gestation and point towards potential areas of research to improve delivery room care for these infants.

The review is comprehensive and precise, but require some corrections before accepted.

  1. Line 103: Replace ECC with DCC.
  2. Line 31: Change '23+6' to '24+6' weeks of gestation
  3. Line 62-63: Please replace 'importantly,' with 'important'
  4. Lines 140-141: Here the authors mention that caffeine in DR improves diaphragmatic contractility and tidal volume and cite reference 43, however, this study does not specify that the caffeine was administered in the delivery room. Please correct this statement to reflect the same. for example: Caffeine administration in the delivery room was associated with an increase in diaphragmatic activity and tidal volume within five minutes in infants between 26 to 34 weeks of gestation. Or you could delete this statement and reference altogether.
  5. Lines 146-153: I would suggest incorporating this paragraph to section 4.3 (Oxygen titration) to provide a flow to the readers.
  6. Line 168: Replace 'extreme' with 'extremely'
  7. Line 250: Replace 'it reduces' to 'they reduce' as the authors are talking about 'higher peep levels' which is plural.
  8. Line 278: Reframe the sentence as 'INSURE comprises of endotracheal intubation and subsequent.......early extubation'
  9. Lines 300-304. Please delete 'similar' as the results of the VON study are not similar to the studies reported previously. Would also replace the word 'require' on line 302 with 'receive' as the statement falsely insinuates that infants who did not require DR-CPR did better compared to those who required CPR. 
  10. I would suggest the authors to include a section on temperature regulation as part of delivery room care in these infants.

Author Response

We would like to thank the reviewers for their thoughtful comments. We have used them to improve the presentation of our manuscript!

REVIEWER 1

The authors provide a review of the available evidence on resuscitating preterm infants <25 weeks gestation and point towards potential areas of research to improve delivery room care for these infants.

The review is comprehensive and precise, but require some corrections before accepted.

  1. Line 103: Replace ECC with DCC.

We decided not to replace ECC with DCC. The study by Ram Mohan et al (Resuscitation 2018, 130, 88-91; reference 29) compared C-UCM with ECC not DCC. This single-center study of 60 infants found that C-UCM increase peak hematocrit concentrations within 24 hours after birth. This can also be seen at Table 4 of the study “Umbilical Cord Management for Newborns <34 Weeks’ Gestation: A Meta-analysis” by Seidler et al. (Pediatrics 2021, 147; reference 21). No studies were identified for any of these comparisons DCC versus C-UCM or I-UCM versus C-UCM.

  1. Line 31: Change '23+6' to '24+6' weeks of gestation

We changed the line according to the reviewer’s suggestion.

  1. Line 62-63: Please replace 'importantly,' with 'important'

We changed the lines according to the reviewer’s suggestion.         

  1. Lines 140-141: Here the authors mention that caffeine in DR improves diaphragmatic contractility and tidal volume and cite reference 43, however, this study does not specify that the caffeine was administered in the delivery room. Please correct this statement to reflect the same. for example: Caffeine administration in the delivery room was associated with an increase in diaphragmatic activity and tidal volume within five minutes. Or you could delete this statement and reference altogether.

We changed the wording to:

“Caffeine administration was associated with an increased diaphragmatic activity and tidal volume breathing within five minutes of its administration in infants between 26 to 34 weeks’ gestation”

  1. Lines 146-153: I would suggest incorporating this paragraph to section 4.3 (Oxygen titration) to provide a flow to the readers.

We decided to not incorporate this paragraph to section 5.3. (former 4.3.), since respiratory drive and spontaneous breathing are discussed separately in this chapter (5.1.) and we feel that oxygen should also have its place in this section, since it might be an important factor to trigger spontaneous breathing. Nevertheless we are now referring to chapter 5.3.

“However, this strategy requires further studies as current neonatal resuscitation guidelines recommend low initial FiO2 as discussed in chapter 5.3. Oxygen titration [4-7].”

  1. Line 168: Replace 'extreme' with 'extremely'

We changed the line according to the reviewer’s suggestion.

  1. Line 250: Replace 'it reduces' to 'they reduce' as the authors are talking about 'higher peep levels' which is plural.

We changed the line according to the reviewer’s suggestion.

  1. Line 278: Reframe the sentence as 'INSURE comprises of endotracheal intubation and subsequent.......early extubation'

We changed the lines according to the reviewer’s suggestion.

  1. Lines 300-304. Please delete 'similar' as the results of the VON study are not similar to the studies reported previously. Would also replace the word 'require' on line 302 with 'receive' as the statement falsely insinuates that infants who did not require DR-CPR did better compared to those who required CPR. 

We changed the lines according to the reviewer’s suggestion.

  1. I would suggest the authors to include a section on temperature regulation as part of delivery room care in these infants.

We included a section about temperature control:

Thermal care to maintain the body temperature between 36.5 and 37.5 °C is central to reduce morbidity and mortality in premature infants [37,38]. The admission temperature has been demonstrated to be a strong prognostic factor in low birth weight infants [39,40]. Placed under a preheated radiant heater, the premature infant should be completely covered with polyethylene wrapping (apart from the face) without prior drying [6,7,37,38]. The temperature should be regularly monitored after birth to prevent hypo- and hyperthermia [6]. Further interventions might be beneficial including the use of warmed humidified gases in infants receiving respiratory support, increased room temperature >25°C, warm dry blankets, thermal mattress, and head cap [6,7,37,38,41,42].

Reviewer 2 Report

I enclose my comments on a separate file.

Author Response

We would like to thank the reviewers for their thoughtful comments. We have used them to improve the presentation of our manuscript!
